# Nanostructure Analysis of Anodic Films Formed on Aluminum-Focusing on the Effects of Electric Field Strength and Electrolyte Anions

**DOI:** 10.3390/molecules26237270

**Published:** 2021-11-30

**Authors:** Sachiko Ono

**Affiliations:** Department of Applied Chemistry, School of Advanced Engineering, Kogakuin University, 2665-1 Nakano, Hachioji, Tokyo 192-0015, Japan; sachiono@cc.kogakuin.ac.jp

**Keywords:** anodic film, aluminum, nanostructure, electric field strength, anion incorporation, defect, electrolyte type, sealing, self-ordering, α-alumina membrane

## Abstract

In this review, the research conducted by the authors on anodic oxide films on aluminum is described, paying particular attention to how the electric field strength, as a factor other than voltage, controls the nanostructures and properties of the films. It will also be indicated what factors contribute to the formation of defects, which, in contrast to the ideal or model film structure, contains a significant number of defects in the film. In addition to electrochemical measurements, the films were examined with a variety of advanced instruments, including electron microscopes, to confirm the “reality of film nanostructure” from a slightly different angle than the conventional view. The following topics on anodic films formed in four types of major anodizing electrolytes are discussed: pore initiation process, steady-state porous structure, sealing mechanism, the relationship between cell parameters and voltage/electric field strength, amount and depth of anion incorporation, electrolyte types, radial branching of pores, atypical pore structures, defect formation mechanism, self-ordering, Al coordination number, and the creation of α-alumina membranes.

## 1. Introduction

Anodic oxide film has a history of nearly 100 years since its development [1,2], and its unique nanostructure of porous oxide with a hexagonal array of pores has attracted the interest of many engineers and researchers in both industrial applications and academic basic research fields. Anodic oxide films are well-known to have evolved from their initial development for corrosion resistance and decorative applications, and functional applications of their nanostructures have been highly anticipated in recent years [3,4]. Specific applications that take advantage of its corrosion resistance and decorative properties include surface treatment of aluminum sashes, housings for electrical equipment, aluminum alloys for automobile engines, and the inner surface of vacuum chambers. In addition to filters, catalyst supports, and photonic crystals, the nano-porous structure of an anodic film can be used as a template for the creation of nanomaterials for electronic, magnetic, and optical devices, including batteries. In recent years, the creation of antireflective coatings has become a practical application. The dielectric properties of barrier-type anodic alumina are widely used as dielectrics in electrolytic capacitors and as gate insulators.

The presentation of the cylinder structure model by Keller et al. in 1953 [5], the clarification of the voltage dependence of cell parameters by J. P. O’Sullivan et al. in 1970 [6], and the discovery of the self-organized condition of the cell arrangement by H. Masuda et al. in 1995 [7] had a significant impact on the subsequent development and application of anodic oxide films, as well as basic research. The recent successful formation of porous films by anodic oxidation of titanium was groundbreaking, allowing the formation of porous films on many metals and alloys [8,9,10] that were previously thought to form only barrier-type films. This breakthrough broadened the field of application of anodic oxide films and greatly increased their attention. Other challenging, innovative, and useful technologies have been developed tirelessly over the past 100 years, and today, the high functionality of anodic oxide films is widely used as an indispensable technology in both fundamental and advanced fields. However, the basic mechanism of anodic film growth is not fully elucidated, even though numerous studies on the technology directly necessary for its practical use, such as the fabrication of highly ordered films, have been undertaken [11]. The creation of oxide films with new functions cannot be achieved without an understanding of the basic formation mechanism, including the effects of electric fields on film nanostructure and anion incorporation. For example, to develop methods to produce high-quality anodic oxide films with lower energy costs, a deeper understanding of the mechanisms of film growth is needed, and the currently accepted knowledge may not be sufficient.

Through long-term studies of anodic oxide films, the authors realized that direct observation of the films by electron microscopy is particularly important in addition to steady electrochemical measurements [12,13,14,15,16]. As a result, we began to concentrate on the real image, which includes the local structure of the anode oxide film that is hidden in the idealized model structure [16]. Differences in the nanostructure and anion incorporation behavior of the films were caused by differences in the electrolyte type and current density, and the importance of the electric field strength in controlling these differences was demonstrated [15,16].

Electron microscopy, whether it is based on the conventional method with an insufficient resolution or the latest electron microscopy with high resolution using today’s advanced sample preparation techniques, has the advantage of direct observation that can be identified the differences in local structure compared to the average evaluation of film structure when using indirect methods. Modeling is important, because it allows researchers to advance by universalizing the subject, but it has the disadvantage of being easy to regard as perfect. The presence of defects that are ignored in modeling, as well as the factors that cause them, appeared to be an important and fascinating issue. The author has also studied the structure and properties of barrier-type films as a basis for elucidating the formation behavior of porous films [17]. As a result, the author realized that not only the geometric structure of the film but also the anion incorporation has a significant impact on the film’s dielectric properties [17,18,19].

This review focuses on the research conducted by the authors, especially on the effects of electrolyte anion incorporation and electric field strength on the nanostructure of anodic oxide films. Due to space limitations, the results were limited to high-purity aluminum (99.99%), and the majority of the applied studies and detailed analyses were omitted. However, I hope that the results of the basic analyses presented here will lead to the reader’s research and technology being further deepened and developed.

## 2. Pore Initiation Process and Porous Structure at Steady State

### 2.1. TEM Analysis of Pore Initiation Process

A lot of studies have been conducted on anodic films, and currently, the following views are accepted regarding the morphology of the pores [3,4,6,13,14,20]. The pore diameter in the steady state is proportional to the formation voltage (about 1 nmV^−1^) and is approximately one-third of the cell diameter (porosity: 0.1 [14]). Many small pores are formed in the entire oxidized Al during the initial stage of film formation, preferentially in the concaved part of the initial oxide [20]. Eventually, the size of the pores becomes proportional to the voltage, and the pores grow in a straight tubular shape perpendicular to the substrate.

Here, the initial pore size being much smaller than the steady-state pore size corresponding to the voltage is discussed in relation to the electric field strength using the direct observation of the cross section by electron microscopy. It is known that the barrier film forms first during the early stages of anodizing and that small pores form later [6,16,20,21,22,23,24,25,26,27]. However, studies on pore generation have mainly been conducted on electropolished Al substrates, and the hemispherical depression of the substrate caused by electropolishing, where size is dependent on the polishing voltage, has a significant effect on the form of the initially generated pores [24]. At the early stages of anodizing, film thickening on the substrate ridge and the resulting concave formation, which led to the formation of initial pores, was reported [20,21]. Therefore, in this experiment, the Al substrate was planarized to less than 1 nm [16,24] before anodizing to remove surface irregularities caused by electropolishing.

Figure 1a–e shows TEM images of vertical cross-sections of anodic films formed in 0.3-mol·dm^−3^ oxalic acid at 40 V for 2.5, 10, 30, 60, and 120 s, respectively. Each time point is presented on the I–t curve. The film thickness of the flat barrier film and the anodizing ratios were 43 nm and 1.1 nmV^−1^ (A), respectively, when the current was decreasing (Figure 1a). Circular spots observed inside the film were caused by electron beam damage, presumably due to the incorporated anions. The barrier oxide thickness was approximately 56 nm during the lowest-current period (Figure 1b), and the anodizing ratio was 1.4 nm·V^−1^ (B); thus, the electric field strength during barrier film growth was 7.2 × 10^8^ V·m^−1^. In addition, the anodizing ratio and electric field strength during the initial small pore growth (C) were 1.1 nm·V^−1^ and 9.1 × 10^8^ V·m^−1^, respectively. For the main pore growth (D), the anodizing ratio and electric field strength were 0.96 nm·V^−1^ and 10 × 10^8^ V·m^−1^, respectively (Figure 1c). When the current density reached a steady-state value after 60 s (Figure 1d), the anodizing ratio and electric field strength were 1.1 nm·V^−1^ and 9.1 × 10^8^ V·m^−1^, respectively (F).

These results indicate that the pore diameter decreases to less than that of the normal main pore generation of approximately 10 × 10^8^ V·m^−1^ when pores are formed by dissolution under a low electric field strength. In addition, the occurrence of radial nanobranched pores is observed during the current recovery period [28,29] when the voltage suddenly drops. In this case as well, it can be stated that the decrease in the electric field strength causes the nanopore generation. Flow instability caused by spatially nonuniform compressive stress [25], mechanical stress [26], and field-induced instability [27] has been proposed as the cause of small pore initiation. Although these mechanisms may contribute, we have confirmed that the main factor determining the small size of the initial pores is the low electric field strength.

In Figure 1d,e, TEM images of cross-sections of anodic films formed for 60 and 120 s showed radially branched nanopores in the center of the cells in both cases, even in the self-ordering anodizing condition of oxalic acid at 40 V. Furthermore, when we verified the STEM images of the film formed in 1.5-mol dm^−3^ sulfuric acid at 20 V, we discovered that radially branched nanopores were frequently generated in the main pores [16]. Radial nanobranched pores, described as “feather-like”, are commonly observed in anodic films formed in chromic acid [30,31] and sodium borate solution [32,33], but such pore shapes have not been reported in the literature for oxalic or sulfuric acid films. In an exceptional case, a sulfuric acid film grown in a confined space was shown to have a periodic dendritic inner pore structure [34]. This was explained by the intermittent flow of volume-expanded oxide into the pores, but it may have been related to radial pore branching, which occurs under low electric fields. In any case, we have long observed the presence of protruding substances near the bottom of the pores in the SEM image of the film formed in oxalic acid, even after a thorough cleaning. They were somewhat strange, and for a long time, I could not figure out why. Due to their fineness, which is difficult to detect in conventional SEM images, and because they quickly disappear due to chemical dissolution in the electrolyte, these radial nanopores within the main pores have been overlooked. These results thus indicate that the formation of radial nanobranched pores is not unique to chromic acid films. Instead, this is a universal phenomenon of anodic films that frequently occurs when the electric field strength decreases during anodizing, since we have also detected radially branched nanopores in the STEM images of the film formed in 1.5-mol·dm^−3^ sulfuric acid at 20 V [16]. In 1978, Thompson et al. [35] reported the first horizontal cross-section observations of various porous anodic films, stating that “steady-state anodizing is not as regular as previously assumed”. This is still true, and understanding the actual structure of the porous film is critical for a more effective application of the film.

### 2.2. Schematic of Pore Generation and Growth Process

Based on the above TEM observation of cross-sections [16], as well as AFM observation [24], schematic representation of the growth process of porous anodic film at different stages is demonstrated in Figure 2. This differs from conventional theory, because when using a nano-flattened Al substrate, as confirmed by our direct TEM observations of the cross-sections, small pores were uniformly generated in the very early stage of the anodic oxidation process.

### 2.3. TEM Analysis of Porous Structure at a Steady State

To elucidate the details of the film structure, TEM was used to observe anodic films while preparing cross-sections with an ion slicer. Figure 3 illustrates TEM images of anodic films formed in (a) 0.3-mol dm^–3^ oxalic acid at 40 V for 3 min and (b) 1.5-mol dm^–3^ sulfuric acid at 20 V for 1 min [36]. As shown in Figure 3a, the TEM image of the film sample indicates small pores near the surface, large pores in a main porous oxide layer, and an underlying barrier layer. The diameter of the steady-state pores was 36 ± 4 nm, and the thickness of the barrier layer at the bottom was 40 ± 2 nm. A bottleneck-like pore shape near the surface, growth termination of the initial small pores, and radial pore branching in places as indicated by arrows were observed. Scattered white, circular spots ranging in diameter from approximately 5 to 20 nm in the pore region must be due to damage caused by electron beam irradiation leading to OH/anion transpiration and re-arrangement of the oxide lattice [35,37,38].

Although cell and pore diameters are small, depending on the formation voltage of 20 V in the case of a cross-section of the film formed in sulfuric acid (Figure 3b), similar pore size changes near the surface region and radial pore branching were frequently observed, as indicated by arrows. As a result, radial nanobranching of the main pores occurred in both films formed, even in conventional major electrolytes such as oxalic acid and sulfuric acid, where the pores are thought to grow columnar. Therefore, we proposed a new perspective at the pores of the anodic oxide film: when the electric field strength decreases, the pores of the anodic oxide film grow in a radially branching manner [16]. This is the cause of the pore wall unevenness and protrusions, and it was probably overlooked until now, because it is easily removed through chemical dissolution in the electrolyte.

## 3. Mechanism of Hot Water Sealing

### 3.1. Sealing of Anodic Films

The sealing process plays an important role in expanding the industrial demand for aluminum, and most anodized aluminum is sealed to ensure corrosion resistance. In 1929, Setoh and Miyata [39,40] reported a sealing method in which the pores were plugged using pressurized steam, dramatically improving the corrosion resistance of the film. Since then, various sealing methods have been developed, including boiling water sealing (also known as hot water sealing or hydrothermal sealing) and sealing with metal salts or other compounds (Ni-based sealing, Cr(VI)-based sealing, Li-based sealing, Ce-based sealing, NaAlO_2_ sealing, organic acid sealing, and so on) [41]. At present, the sealing method using hot water is mainly used, but room temperature sealing is also being considered from the viewpoint of energy saving [42]. 

The dissolution of anodized alumina in the pores, followed by the precipitation of boehmite or pseudo-boehmite, has now been perceived as the possible principle for the sealing process in hot water [41,42,43,44,45,46,47,48,49,50,51,52,53,54,55]. The density of the hydrous oxides is reported to be 2.6–2.7 [50], whereas the original anodic oxide density is 2.9–3.4 [50,56], resulting in volume expansion and pore plugging. Wefers [47,48] summarized the sealing process as follows: the pore wall material reacts with water to form gelatinous boehmite and amorphous alumina hydroxide. The gelatinous boehmite that forms on the pore walls condenses to form pseudo-boehmite, which recrystallizes to form the final boehmite. During the final stage of the sealing process, he also reported the formation of an “intermediate layer” beneath the outer surface. Using impedance measurements, Hoar and Wood [43] suggested preferential pore plugging at the outer part of the film and gradual narrowing of the pores via boehmite formation. Diggle et al. [44] supported the above-mentioned mechanism for gel-type boehmite precipitation, followed by condensation and crystallization and early pore mouth closure, as observed in the majority of other sealing studies. However, the details of these phases’ morphologies and the actual formation process of the layered structure of the sealed film remain insufficiently revealed.

### 3.2. TEM Analysis of Sealing Process

In recent years, the performance of SEM and TEM has dramatically improved, and the technique for preparing observation samples such as cross-sections has significantly improved, allowing direct and detailed observations of previously unimaginable film structures. Therefore, we focused on the observation and characterization of structural change in the anodic porous films with hot water sealing, especially the cross-sections, using these advanced analyzers to obtain in-depth information. We aimed to clarify the mechanism of hot water sealing in particular, paying attention to the temporal change during the reaction in the sealing process, differences in the local structure of the film, and differences in the sealing process due to the different incorporated anions from anodizing electrolytes [36].

Figure 4 shows TEM images of cross-sections of anodic films formed in (a,b) 0.3-mol dm^−3^ oxalic acid at 40 V for 3 min and (c) 1.5-mol dm^−3^ sulfuric acid at 20 V for 1 min (a) before and (b,c) after sealing in boiling water for 10 min [36]. When immersed in boiling water, the film dissolves, and boehmite precipitates as small flakes as a result of (100)-oriented growth, dispersing in the pores and on the surface. The number and size of the flakes increase as the sealing time increases while maintaining flake thickness, resulting in a densely intertwined structure. Eventually, a three-layered structure is formed, composed of an outer coarse flaky boehmite layer, an intermediate densely intertwined boehmite layer, and a porous layer whose pores are filled with dense boehmite. A schematic of the sealing process in hot water is shown in Figure 4d.

### 3.3. Sealing Mechanism

The detailed process of hot water sealing and the nanostructure of anodic oxide films formed in various electrolytes were studied further using direct TEM/SEM observation and advanced sample preparation methods, as well as a XPS depth analysis, yielding the following results [36]. 

At the initial stage of sealing (1–1.5 min), the pore walls and the surface of the film dissolves upon immersion in boiling water, and the hydrated alumina/boehmite immediately precipitates as small flakes evenly dispersed in the pores and on the top surface. These precipitates are not gelatinous and have a diameter of 10–20 nm and a thickness of 3 nm. When the anodic film is several μm thick, ion outward diffusion is restricted, and precipitation begins preferentially near the pore bottoms. The flaky precipitates are boehmite, according to the electron diffraction patterns and high-resolution lattice images [36]. 

Akahori et al. [45] reported that the spacing of the (020) plane of a sealed anodic film is in the range of 0.62–0.63 nm, which is larger than 0.611 nm for standard boehmite. This discrepancy is due to excess water; that is, more than one water molecule for one alumina molecule, and the presence of excess water in the interlayers of the boehmite structure. While these low crystallinity hydrated oxides formed after sealing of anodic film are often called pseudoboehmite, it has been established that the crystal structures of pseudoboehmite and boehmite are not significantly different [57]. Okada et al. [58] found that small-crystallite boehmite, called pseudoboehmite, showed broadened X-ray peak widths and increased (020) reflection d-spacing. These changes were attributed to the crystallite size, corresponding to smaller numbers of (020) stacking layers and excess water in the interlayers of the boehmite structure.

Due to the flake thickness of approximately 3 nm suggesting five stacking layers of (020), the composition is estimated to be generally Al_2_O_3_·1.3H_2_O [36]. Precipitates take on platelet/flake-like shapes due to the strong (100)-oriented growth of boehmite that forms thin (020)-layered structures. The boehmite precipitates are flaky rather than acicular; however, the flakes are frequently rolled up at the edges, resulting in an apparent needle-like/acicular shape. 

At the intermediate sealing stage (1.5–3 min), the quantity and size of flaky boehmite increase with the increasing sealing time while maintaining a flake thickness of around 3 nm, corresponding to several layers of (020) stacking, eventually forming intimately intertwined flaky boehmite [36]. During this stage, the pore size of the anodic film expands from its original diameter, and the barrier layer becomes thinner due to dissolution. After the pores are filled with intimately intertwined flaky boehmite, and the dissolution/hydration reaction inside the pores is completed; a large flaky boehmite on the top surface begins to grow outward in a vertical direction.

Thus, a three-layered structure is formed (3 min of sealing or later), consisting of an outer coarse large flaky boehmite layer, an intermediate densely intertwined flaky boehmite layer, and a hydrated porous layer whose pores are filled with densely intertwined flaky boehmite [36]. With increasing sealing time, the top two hydrated layers having a roughly 2:1 thickness ratio become thicker, while the porous layer becomes thinner. The combined thickness of the intermediate layer plus the porous layer is almost equal to the total thickness of the original film. The intermediate dense layer is assumed to grow in a more concentrated environment of hydrated alumina, whereas the outer rough flake layer is assumed to grow vertically from the root to the tip in an outward diffusing environment of less concentrated hydrated alumina. 

The volume expansion after hot water sealing by changing anodic alumina to boehmite (Al_2_O_3_·1.3H_2_O) is 1.4 times on the calculation [36]. On the other hand, the deposited boehmite is not compact but flaky, and the actual filling rate in the pores and the intermediate layer is estimated to be about 62%. After sealing, the sulfur content in the porous layer of the anodic film formed in sulfuric acid decreases from 1.7 to 1.6%. However, the sulfur contents in the intermediate, and outer layers are estimated to be 0.25% and 0.04%, respectively, indicating that most anions diffuse out through the pores. 

The sealing behavior of the films formed in phosphoric acid and chromic acid are similar to those of the films formed in oxalic acid and sulfuric acid, producing a three-layered hydrated structure [36]. Notwithstanding, the sealing speed calculated from the time required for pore filling is greatest in the sulfuric acid-formed film and lowest in the phosphoric acid-formed film. Given their anion content/dissolution rate, the thickness of the intermediate layer of the phosphoric acid film is unexpectedly low, whereas the thickness of the oxalic acid film is unexpectedly high. It is deduced that anodic oxide solubility in hot water differs characteristically from that in acid solution, presumably owing to the complex-forming ability of each anion with aluminum. Phosphate aluminum complexes suppress anodic film dissolution in hot water, while oxalate aluminum complexes assist the same. Pore diameter and anodic oxide solubility in hot water, as determined by the content and complexing ability of incorporated electrolyte anions in the films, would both influence the sealing speed [36].

## 4. Cell Parameters of Anodic Films Formed in Four Typical Electrolytes

### 4.1. TEM Images of the Films Formed at 20 V in Four Anodizing Electrolytes

There have been various attempts to apply porous anodic films as functional materials, such as membranes and molds for producing metallic or semiconductive porous films. The ability to precisely control the structure at the nanometer scale is the most important key to practical applications in these cases. We used electron microscopy and electrochemical measurements to investigate cell parameters such as the cell diameter, pore diameter, and barrier layer thickness of porous films produced in four different anodizing electrolytes [14,15]. These cell parameters are conventionally interpreted as being proportional to the voltage, but here, we pay more attention to the relationship with the electric field strength, namely current density, and analyze them as a function of it.

The characteristics of the four typical anodic oxide films formed in each type of electrolyte are as follows. Sulfuric acid, which is inexpensive and the most widely used electrolyte in the anodizing industry, produces a transparent film. Sulfuric acid films are used not only for corrosion resistance and decorative purposes but also for templates when a smaller size is required due to its low formation voltage of around 20 V, which also lowers energy costs. Oxalic acid film has a formation voltage of around 40 V and is used when a higher corrosion resistance is required. The film is transparent but slightly yellow. Chromic acid film is an opaque grayish-white in color and has a ceramic-like appearance. The anodizing process is carried out at approximately 60 V or higher, and it is mainly used for Al alloys for aircraft. A phosphoric acid film is formed at relatively high voltages of 80 V or more, has a whitish appearance, and is characteristically difficult to hydrate. It is also used as a base for painting.

Figure 5 shows TEM images of stripped anodic films formed at 20 V in (a) 1-mol dm^−3^ sulfuric acid at 20 °C, (b) 0.3-mol dm^−3^ oxalic acid at 30 °C, (c) 0.3-mol dm^−3^ chromic acid at 40 °C, and (d) 0.4-mol dm^−3^ phosphoric acid at 25 °C [59,60]. The pore size varied greatly depending on the electrolyte in the order of sulfuric acid, oxalic acid, chromic acid, and phosphoric acid, even though the anodizing voltage was the same. The average pore size of these films was more than twice as large for phosphoric acid as for sulfuric acid [14]. The relationship between pore size and electrolytic conditions has been studied extensively but mostly for a single electrolyte. As a result, such a difference in pore size depending on the electrolyte is not well-understood.

### 4.2. Current Density—Voltage Diagram

The steady-state current values after 10 min of electrolysis at a constant voltage in each electrolyte are shown in Figure 6 to investigate the cause of these differences in pore sizes. As shown in the graph, the steady-state current increases exponentially with the increasing voltage in all electrolytes and rapidly when a certain threshold is exceeded, resulting in the dielectric breakdown (burning). We found that the highest electrolysis voltage that avoided burning (i.e., local concentration of the current) was the self-ordering condition [61,62,63]. The porosity of all the films prepared under these self-ordering conditions in sulfuric acid, oxalic acid, and phosphoric acid was 0.1, even though the formation voltage differed [61]. The dotted red ellipses indicate the most used voltages for each electrolyte, but the current values there vary greatly, depending on the electrolyte. The current value in 20-V anodizing decreases in the following order: sulfuric acid > oxalic acid > chromic acid > phosphoric acid.

As will be discussed later, the high field mechanism of anodic film growth suggests that the log of the current density has a linear relationship with the electric field strength [64]. As a result, a higher current density implies a higher electric field strength in the barrier layer during anodizing at the same voltage. The large difference in the pore size of the films formed in the four electrolytes at the same voltage is thought to be due to a difference in the electric field strength during anodizing [14,15].

### 4.3. High Field Mechanism

Güntherschltze and Betz [64] empirically found the following equation to hold for the barrier film formation on various valve metals:*j* = A_1_ exp (β*E*) i.e., *E* = A_2_ log *j* (modified by this author)
where *j*: current density; *E*: electric field; and A_1_, A_2_, and β: constants.

This equation was theoretically verified by many workers on the anodizing of various metals, which were called high field mechanisms [65,66,67]. The film thickness of each metal is inversely proportional to the logarithm of the ionic current when the film is formed up to the same voltage, according to the above equation of ionic conduction at a high field strength for the anodic barrier film grown on various metals. Thus, it is indicated that the log of the current density log *j* is proportional to the electric field strength *E*, i.e., the formation voltage/film thickness ratio at the barrier layer. Dell’Oca and Fleming [67] certified the above equation using ellipsometry for the growth of the initial pore-free films formed on aluminum in phosphoric acid, which is a typical pore-forming electrolyte.

### 4.4. Cell Parameters

Cell parameters of anodic films formed at a constant voltage up to 120 V in four typical electrolytes are shown in Figure 7 [14,15]. Note that Ono and Masuko [14,15] only presented results up to a formation voltage of 40 V, but here, results up to 120 V were newly determined. When a small current of 1 A m^−2^ was applied to the sample in a neutral borate solution, the voltage jump value was used to calculate the thickness of the barrier layer as the film resistance [14]. Under the conditions of the current experiment, a voltage jump of 1 V corresponded to a barrier layer thickness of 11.4 nm based on the results of the direct TEM observation of the barrier layer. As shown in Figure 7a, the thickness of the barrier layer increased almost linearly with the increasing voltage in all electrolytes. The inserted dotted line corresponds to 1 nm/V. 

The thickness of the barrier layer was not the same for the four electrolytes but was larger for sulfuric acid, oxalic acid, phosphoric acid, and chromic acid, in that order. As shown in Figure 7b, the average value of the cell diameter obtained by TEM observation increased almost linearly with the increasing voltage up to about 60 V, but the voltage ratio became larger above that level. The inserted dotted line corresponds to 2.5 nmV^−1^, indicating that the cell diameter is approximated to be 2.5 nmV^−1^. The cell diameters increased in the same order as the barrier layer thickness for sulfuric acid, oxalic acid, phosphoric acid, and chromic acid, but the difference due to the electrolyte was smaller than for the case of the barrier layer thickness.

Next, the pore diameter was calculated from the porosity α (Figure 8) and cell diameter: pore diameter = √α·cell diameter. As shown in Figure 7c, the pore size increased with the increasing voltage above 5 V. However, below 5 V, the pore diameter increased with the decreasing voltage. This unusual change in pore size below 5 V can be seen in the TEM images inserted at the right end of Figure 7c, which is presumably due to a low electric field [15]. Sulfuric acid, oxalic acid, chromic acid, and phosphoric acid have larger pore sizes at the same voltage, in that order. In these figures, it should be noted that the difference in pore size between the films formed in the four electrolytes is significantly larger than the difference due to the electrolyte in the thickness of the barrier layer and the diameter of the cell.

Using these cell parameters, the cell-to-pore diameter ratio (*d*cell/*d*pore) of the anode films formed in both sulfuric and oxalic acids at different constant voltages was plotted against the logarithm of the current density (log *j*), and a linear relationship was found [15]. Since the electric field *E* across the barrier layer of the porous film varied linearly with log *j*, as in the case of the barrier-type film [17,19], the *d*cell/*d*pore ratio was considered to be proportional to the electric field. From these results, it was deduced that the barrier layer thickness and cell diameter were proportional to the voltage and inversely proportional to *E*, whereas the pore diameter was proportional to the voltage and inversely proportional to the square of *E*. This result clearly shows that the pore size depends more strongly on the current density than on the other cell parameters. The sulfate anion content of the films formed in sulfuric acid increased linearly with log *j* (i.e., increase in electric field *E*) [15]. Recently, Minguez-Bacho et al. reported by Rutherford backscattering spectroscopy that the incorporation of sulfur species into anodic aluminum oxide and the rate of volume expansion showed a logarithmic dependence on the average current density, regardless of the applied voltage or sulfuric acid concentration [68]. Thus, the electric field strength is the most important factor governing the structure and properties of anodic oxide films.

### 4.5. Porosity

Figure 8 shows the change in porosity with voltage for the films formed in four different electrolytes [14,61]. Porosity was measured using the pore-filling method (re-anodizing). That is, the method is to apply a small current—5 A m^−2^, in this case—to a sample in a neutral borate solution and determine the porosity from the gradient of the voltage rise. For details, please refer to other reports [14,61]. The porosity increased with sulfuric acid, oxalic acid, chromic acid, and phosphoric acid and decreased with the increasing voltage. It can be seen that, when the formation voltage is less than 10 V, the porosity increases significantly, as also shown in the TEM images inserted in Figure 7. The ratio of the pore diameter to the cell diameter of the film with a formation voltage of 20 V is larger than that of the film with a formation voltage of 5 V. This means that the reduction of the pore diameter is smaller than the reduction of the cell diameter due to the voltage decrease. It should be noted that, as the voltage decreases, so does the current, and the electric field strength in anodic oxidation eventually decreases in the low voltage range.

## 5. Incorporation Depth and Content of Electrolyte Anions

### 5.1. TEM Images of Horizontal Cross Sections of the Films Formed in Four Typical Electrolytes

Thompson et al. [35] showed that the ion milling method can be applied to the preparation of horizontal cross-sections of porous films and that TEM observations can provide a great deal of information on the microstructure of the films. We prepared horizontal cross-sections of films formed by ion milling in four different electrolytes and directly observed the shape of the cells/pores and the state of the anion-free layer at the cell boundaries [12,13,16,30,60,69,70]. Electrolyte anions are drawn into the barrier layer from the electrolyte under high electric fields. Therefore, the higher the electric field and the larger the negative charge of the anions in the oxide film, the smaller their size, and the smaller their interaction with the oxide, the greater the number of anions incorporated deeper into the barrier layer. The anion content of porous anodic films linearly increases with the increasing log of the current density, i.e., electric field strength [15], similar to the case of barrier-type films [17,19,59]. It is shown that the anion content and species significantly affect the film properties [23,36,68,71,72,73]. In barrier-type films, the anion species and the content of anions have a significant impact on the permittivity, as well as the leakage current, due to the formation of defects [17,18,19,74,75].

Figure 9 shows TEM images of horizontal cross-sections of films formed at standard anodizing conditions in (a) 1-mol dm^−3^ sulfuric acid at 25 V [69], (b) 0.3-mol dm^−3^ oxalic acid at 40 V [60], (c) 0.4-mol dm^−3^ phosphoric acid at 80 V [13], and (d) 0.3-mol dm^−3^ chromic acid at 80 V [16]. In the film formed in sulfuric acid (a), a circular or elliptical hole exists in the center of the hexagonal cell. When the cells were not regular hexagons, the pores were not perfect circles but, rather, ovals that mimicked the shapes of the cells. The cell boundaries had a white contrast, indicating that anions had most likely penetrated up to the cell boundaries and that cell bonding at the boundaries was weak. This was consistent with the observation that, when the membrane was cracked, only films formed in sulfuric acid delaminated between cells and exhibited a nanotubular structure.

On the other hand, in the film formed in oxalic acid (b), an anion-free layer was observed, with a dark contrast between the cells, and the thickness ratio was about 10%. This film was formed at 40 V under so-called self-ordering condition, but because the film was still thin and not yet fully self-organized, pore branching and the formation of new pores were frequently observed. The shape of the pores was similar to the shape of the cells, as in the case of the sulfuric acid film. In all triple points of the cells, there were small circular defects (voids) that showed a white contrast (Figure 9b). The triple cell junctions were located above the protrusions of the aluminum substrate in the barrier layer [12,60].

In the film formed in phosphoric acid (c), the thickness ratio of the anion-free layer at the cell boundary was about 25% [12,35,38] and increased with the decreasing formation voltage [38]. In addition to the spherical voids, including particles having dark contrasts at the triple-cell junctions, many crack-like elongated side holes penetrated the cells, and the pores were found [12,13,37]. According to the morphology of the elongated holes, they were formed by the drilling effect of the electronic current caused by the small breakdown. These straight side holes that cross the cell wall almost horizontally are often observed, especially in anodic films formed in phosphoric acid, chromic acid, and Ematal bath [12]. For the first time, a horizontal cross-sectional observation revealed that defects in the porous film were caused by a small breakdown accompanied by gas evolution of the barrier layer between the top of the protruding part of the substrate, which was the triple-cell junction, and the bottom of the pore [12,13,16,73,76].

We previously presented a model in which a microscopic dielectric breakdown created gas-filled spherical voids in the triple points of cells in addition to side holes in the cell walls, as observed by TEM, and these voids were vertically aligned in an almost regular fashion in 1991 [12,60]. Such spherical voids in a phosphoric acid film were also later reported by Le Coz and Arurault [77] and Molchan et al. [78] in 2010. Molchan et al. suggested that such voids in the triple-cell junction were formed by the enrichment of copper impurities and resulting gas generation [78]. Hause and Hebert stated that the prediction of local tensile stresses in nanoscale ridges at the metal–film interface was supported by the observation of voids in these areas [79].

The pore morphology of the film formed in chromic acid appeared to be unique. One major pore grows radially branching into many nanopores. We recently studied the radial nanobranching of the pores in detail [16]. The resulting nanobranching of the pores is a phenomenon that occurs when the solubility of the barrier layer decreases [16,80] and the strength of the electric field decreases, most commonly in the case of chromic acid. However, this phenomenon is not unique to chromic acid and is frequently observed in the films formed in oxalic acid and sulfuric acid [16]. In oxalic acid and sulfuric acid cases, the traces of nano-branching were difficult to detect, because they were quickly lost by dissolution in the electrolyte. However, by using advanced techniques, it was discovered that this phenomenon is not limited to chromic acid but is also observed in the case of oxalic acid (as shown in Figure 1d) and sulfuric acid [16].

The generation of such nanobranched pores is caused by a decrease in the electric field, and this is true for the formation of many small pores in the early stages of film growth or in neutral solutions at high temperatures, such as 80 °C [16]. The shape of radial branching of the pores is due to the curvature of the barrier layer. It is believed that pores of reduced size are formed to promote and sustain the dissolution and growth of the film under low electric fields.

### 5.2. Change in the Thickness Ratio of the Anion-Incorporated Layer with Formation Voltage

Although the existence and ratio of anion-containing and non-anion-containing layers have been investigated using dissolution behavior, chemical analysis, and TEM observations of horizontal cross-sections, it was unclear how the thickness of the layers was affected by the voltage and other electrolysis conditions. As shown in Figure 10 and Table 1, the thickness of the anion incorporation layer increased with the increasing voltage, and at the same time, the amount of anion incorporation increased [38]. This is because higher voltage anodizing results in a higher current density and, therefore, higher electric field strength, which increases the incorporation of anions. These results are in good agreement with the results obtained from the difference in the film dissolution rate measured electrochemically [73], which will be discussed later.

### 5.3. Spherical Voids at the Triple Cell Junctions

Spherical voids with a diameter of 10–30 nm containing dark contrasting particles are always found in the triple-cell junctions of films formed in phosphoric acid, oxalic acid, chromic acid, and Ematal bath where anion-free layers are present [12,13,16,60,73,76]. To clarify the cause of the void formation, a porous film formed in phosphoric acid was prepared. Only the film was dissolved to leave an uneven substrate by dipping the specimen in a phosphoric and chromic acid mixture, and the substrate Al was anodized in a neutral borate solution to grow a barrier-type film [12,76]. As a result, spherical voids containing dark particles were generated in the barrier layer on all the substrate projections, and their sizes increased with the increasing voltage (Figure 11).

Therefore, these voids were formed by gas evolution due to the electronic current flowing on the protrusions of the Al substrate. In addition, crystal lattice images were observed in this particle, which were thought to be alumina crystallized by heating of the electronic current at the time of observation, but later studies revealed that it was a CuAl alloy derived from Cu enriched in the substrate/film interface [16,78]. Figure 11 shows the voids formed on the protrusions and the lattice images of the central particle [76].

## 6. Dissolution Rate and Defect Content Estimated by Pore-Filling Method

Since the anodic film’s porous layer is derived from the barrier layer, the composition and structure of the barrier layer are directly transferred and preserved in the porous film. As a result, clarification of the barrier layer’s structure leads to clarification of the porous structure. We verified the structural changes during the chemical dissolution/sectioning of the films, including both barrier and porous layers, by dipping in sulfuric acid [73]. Figure 12 shows voltage–time curves during re-anodizing of the samples. With the chemical dissolution of the film, the voltage jump value decreased with the decrease in the barrier layer thickness, and the voltage rise gradient decreased with the increase in porosity of the film. When the barrier layer was dissolved to about one-third of its thickness in the phosphoric acid film, no voltage jump was observed, and the voltage rose almost linearly from zero. This indicated that defects in the barrier layer’s center reached the substrate. Such defects, which were detected by TEM observation [12,13,16,60,76], could be quantitatively evaluated by using the pore-filling method. The amount of defect in the films formed in phosphoric acid and chromic acid was significant and increased with the increasing formation voltage [73].

When the voltage jump values are plotted against the dissolution time, the dissolution rate of the barrier layer is greater in the center than near the surface, and the dissolution rate of the inner layer without anion contamination is considerably low (Figure 12b). The dissolution rates of the outer, middle, and inner layers of the films formed in four typical anodizing electrolytes, as well as the formation voltages, are shown in Figure 13.

According to the difference in dissolution rate, the film formed in phosphoric acid has three layers, the film formed in oxalic acid has two layers, the film formed in chromic acid has three layers, and the film formed in sulfuric acid has two layers. The dissolution rate of the inner layer, where no anions were incorporated, was remarkably low. This is why anodizing in pyrophosphoric acid can produce anodic alumina nanofibers; the anion-containing layer is dissolved, leaving only a more insoluble framework consisting of an anion-free inner layer [81]. However, the dissolution rate of the sulfuric acid film’s inner layer was greater than that of the outer layer, indicating that the sulfate anion was concentrated in the inner layer. The high dissolution rate of the middle layer of the chromic acid-formed film is most likely due to the high surface area caused by the presence of many nanobranchings.

The model diagram of the defect structure of the film formed in the four electrolytes, where the hatched parts are anion-free oxides, is shown in Figure 13b, along with the ratio of the anion incorporation depth [12,13,16,60]. The film formed in sulfuric acid has the fewest defects, which is related to a large amount of anion contamination and the absence of a noncontaminated layer. The chromic acid film without anion, on the other hand, is particularly irregular. A relatively large number of defects are observed in the case of a phosphoric acid film where the anion-free layer is approximately 25%. The oxalic acid film has the fewest defects after the sulfuric acid film, but it has small spherical voids [60]. These results suggest that pure alumina tends to induce a dielectric breakdown, and anion contamination probably has the effect of electrically stabilizing the anodic alumina.

## 7. Atypical Pore Structures in Anodic Films

### 7.1. Radial Nanobranching of Pores

Here, we aimed to clarify the original, nonidealized structure and actual growth process of anodic porous alumina film using recently developed observational techniques. Conventionally, it has been assumed that anodic film pores grow in a straight tubular shape [5,6] under field-assisted dissolution [6,44] and/or field-assisted flow mechanisms [79,82,83] and that irregular structures are caused only by unusual states, such as the local current concentration (e.g., dielectric breakdown) [12,80], non-DC components [84], and impurities in the substrate. It is unclear, however, whether this is the case. In the present study, based on the principle that the essence of pore growth and the resultant morphology depends on the electric field strength [12,15], we proposed a universal mechanism for pore growth in most anodized electrolytes [16]. Specifically, we first elucidated the cause of so-called irregular structures, such as pore branching and side holes [16,70,80], when using a chromic acid electrolyte, wherein various atypical structures frequently appeared in the film, and then investigated other electrolytes.

To reveal detailed nanostructures, TEM/STEM images of vertical (Figure 14a) and horizontal cross-sections (Figure 14c,d) prepared via Ion Slicer and FIB, respectively, of the films formed in 0.3-mol·dm^−3^ chromic acid at 80 V (Figure 14a,c,d) were observed. Energy-dispersive X-ray spectroscopy (EDX) and electron diffraction microanalysis (ED) of the horizontal sections were also performed. After the initial small pores in the diameter of around 20 nm grew to a length of about 100 nm in Figure 14a, most of the pores stopped growing, and the cell diameter changed to about 200 nm (Figure 7c), corresponding to an 80-V formation voltage. It can be seen that the morphology of the pores is not straight but branches into smaller pores radially from the central pore. The state of the branching of these nanosized pores is more clearly visible in Figure 14b’s fracture plane SEM image. The radial nanobranching pores also appear to be many projections on the pore wall. The spherical void in the triple point of the cells is indicated by the arrow in this SEM image.

Figure 14c is a TEM image of a horizontal section showing the morphology of the branched pores and the results of EDX analysis of each part of the film. No Cr was detected in the main body of the film, and only a small amount of Cr was detected in the vicinity of the pore wall. The chromic acid film was not contaminated with Cr, as is well-known [31,35,85,86]. The spherical voids and dark contrasting particles encapsulated in the triple-cell junctions are indicated by arrows in Figure 14d, and the EDX and ED analyses revealed that the particles contained Cu, and they were crystalline. Our other study showed that the particles in the voids were CuAl alloy and that Cu enriched in the triple-cell junctions on the protrusions of the substrate was encapsulated as CuAl alloy particles in the voids created by oxygen gas generation due to a small dielectric breakdown and migrated into the film. Such spherical voids and encapsulated crystalline particles also exist in the film formed in phosphoric acid [76], as described in the previous section.

### 7.2. Classification of Various Pore Structures

We summarized a schematic diagram showing various atypical pore structures in Figure 15: (a) small initial pores on the top surface, (b) branching main pore, (c) straight side holes across the cell wall, and spherical voids with CuAl alloy particles at triple-cell junctions, (d) radially branched nanopores within the main pores, and (e) a branching colony structure comprising a bundle of radially branched main pores [13]. The three atypical structures of (b), (c), and (e) were accompanied by dielectric breakdown. Although these structures are typical of chromic acid films, it should be noted that they are not exclusive to chromic acid films and can be found in typical anodic films, depending on the anodic oxidation conditions. In films formed in sulfuric acid and oxalic acid, not only initial micropores and branching but, also, radial nanobranching in main pores can be detected using advanced analytical equipment and avoiding the chemical dissolution of the film [16].

## 8. Self-Ordering Mechanism of Cell Arrangement

In the 1990s, the nanometer-scale fineness, regularity, and controllability of the structure of porous alumina attracted attention, and research for purposes other than providing corrosion resistance and decorative properties began to be actively conducted. One such application is as a perpendicularly magnetized film with metal precipitated in the pores. Furthermore, Masuda et al. [7] discovered the self-organization of pore arrangement with a period of 100 nm in the case of oxalic acid electrolysis at 40 V, which has elevated interest in the functionality of “anodized aluminum” in a variety of fields.

The authors [87] found that similar self-ordering occurred under the condition of hard film formation in sulfuric acid (i.e., high current condition) at around 25 V. In addition, it was reported that self-ordering of phosphoric acid films also occurred at a high voltage of 195 V [88]. The authors discovered, based on the formation behaviors of these ordered films, that the key factor for self-ordering is anodizing under a high current density (i.e., high electric field) just before burning occurs [61,62,63,89]. Figure 16 shows SEM images of the film’s cell array formed under burning. The cell array is highly self-ordered near the center of the burned area where the current is most concentrated, but it is disordered at the periphery. As drawn in the model inserted in Figure 16, self-ordering proceeds during anodic oxidation at a high current density (high electric field). These findings imply that anodizing at a high current density within the range of avoiding burning is a self-ordering condition. Using this principle, self-ordering has been obtained over a wide range of voltages (i.e., a wide range of cell diameters) and with a variety of electrolytes [63].

As for the self-organization mechanism, other proposals have been made, such as the relationship between volume expansion, stress [90], and porosity during oxide formation. According to the findings of the authors’ study, the porosity of the self-ordered films was around 10% regardless of the electrolyte or electrolysis conditions [61]. Further, accelerated film growth leads to increased stress between cells. As a result, geometric morphology, in addition to mechanical stress, is undisputedly related to the self-organization mechanism.

## 9. Coordination Numbers of Al in Anodic Alumina Measured by NMR

Accurate information on the local structure of amorphous alumina obtained by anodic oxidation is useful for a deep understanding of the properties of anodic oxide films and for designing new materials and devices. Many previous nuclear magnetic resonance spectroscopy (NMR) analyses have been reported, but the results have been inconsistent and contentious, most likely because they did not take into account the type and content of incorporated anions. We prepared anion-free and anion-incorporated porous amorphous alumina by anodizing using anion-free chromic acid electrolytes and other typical electrolytes (sulfuric acid, oxalic acid, and phosphoric acid), respectively. The local structure around aluminum atoms was investigated by NMR, and the coordination number of Al to O was studied [91]. As a result, AlO_4_, AlO_5_, and AlO_6_ units were discovered in the structure of anodized amorphous alumina, with AlO_5_ being the dominant unit. The percentages of each unit were 37.7, 54.3, and 8.0 percent, respectively, and the anion-free sample, a chromic acid film, had an average coordination number (N^Al-O^) of 4.70. These results are shown in Table 2, as well as the chemical compositions of the various anodic oxide films obtained from TG-DTA, TG-MASS, and the chemical analysis.

The results of ^1^H−^27^Al cross-polarization and magic-angle spin measurements showed that the percentage of AlO_6_ units decreased significantly when the pysisorbed water was removed by heat treatment to 300 °C. The percentage of oxygen coordination polyhedra (AlO_4_, AlO_5_, and AlO_6_) was closely related to the depth and amount of anion incorporation, as well as the type of anion, and the average N^Al-O^ increased as the anion content increased.

## 10. TG-DTA Studies: Crystallization and Desorption of Anions

As it is well-known, anodic alumina is amorphous, but it releases adsorbed and bound water when the temperature rises due to heating, and it also releases contained anions when it reaches the transition temperature of the alumina and crystallizes. The as-detached film was crushed into a fine powder, and a thermogravimetry-differential thermal analysis (TG-DTA) was carried out [92,93,94]. The properties of the anodic oxide film, as well as the crystallization temperature, and the effect of the incorporated anions can be clarified by examining the weight loss due to the release of the contaminated substances, as well as endothermic and exothermic peaks. Figure 17a shows the TG curves of the films prepared under four typical anodizing conditions, plus anodic oxidation at 25 V in 0.3-mol dm^−3^ sulfuric acid, which is the so-called self-ordering condition, for a total of five films heated in air. The TG-MS curves of the films formed in 0.3-mol dm^−3^ sulfuric acid are shown, as well as the H_2_O released when the films were heated in the He gas environment. In both instances, the heating rate was 10 °C/min.

From the results of TG-MS measurement of H_2_O, we can see that this is certainly due to the release of water, which is probably physically adsorbed on the surface of the film. In the sulfuric acid film, water is again released (less than 1%) around 600–700 °C, which is water or OH that is somewhat tightly bound to the oxide. The chromic acid film with free anions, on the other hand, exhibits no weight loss above 300 °C, indicating that there is almost no moisture present other than through physical adsorption. Even stranger is the phosphoric acid film, which shows no weight change after a slight weight loss of 100 °C. Phosphoric acid is said to be difficult to hydrate [36], but it may have the effect of inhibiting even the physical adsorption of water.

When the temperature was further increased to reach the crystallization temperature of γ-alumina above 800 °C, the weight of the oxalic acid and sulfuric acid films decreased significantly due to the release of anions, as shown in the TG-DTA curve in Figure 17b. However, the weight loss of these films continued gradually up to 1200 °C, close to the transition temperature to α-alumina, suggesting that there was no single state of the anion presence in the films. The first upward peak of TG-DTA was due to the exothermic reaction caused by crystallization in all the films, but the downward peak appeared after the endothermic reaction caused by anion decomposition in oxalic acid and sulfuric acid films. Naturally, no endothermic peak or weight loss was observed in a chromic acid film that did not contain anion, but, strangely, no endothermic peak or weight loss due to anion decomposition was observed in a phosphoric acid film that contained phosphate anion. The desorbed phosphate anion formed aluminum phosphate (AlPO_4_) in the film, which sublimated/evaporated and moved through the film but was not completely released outside the film or remained in the measurement vessel [94]. Therefore, the exothermic peak of AlPO_4_ crystallization was observed at around 1000 °C in the phosphoric acid film.

The transition temperature to γ-alumina is in the order of chromic acid < phosphoric acid < oxalic acid < sulfuric acid, and the lower the anion content, the lower the transition temperature. Except for the phosphoric acid film, the transition temperature to α-alumina is around 1150 °C, because the anions are almost completely removed during the γ-transition, while the phosphoric acid film is around 1300 °C, because AlPO_4_ remains. This information is summarized in the inset table in Figure 17b. These results suggest that not only the geometrical structure but, also, the type and amount of incorporated anion may dominate the film properties related to various functions, such as the dielectric properties of the film.

## 11. Fabrication of α-Al_2_O_3_ Membrane with Tunable Pore Diameters

Anodic porous alumina films, which are typical self-ordered nanoporous materials, have been widely used in a variety of applications, including filters, catalyst supports, and template plates, as well as electronic, magnetic, and optical devices [95,96,97]. However, commercially available anodic porous alumina membranes (e.g., Anodisc^®^ (Maidstone, UK) [98] have insufficient pore arrangements and low chemical resistance, because the porous alumina films are amorphous. To enlarge the application of anodic porous alumina, it is necessary to control the cell dimensions and improve the chemical resistance of alumina membranes in extreme environments, such as high temperatures, high vapor pressure, and high-concentration acid/base solutions. Chang et al. [99] obtained 10-mm diameter α-alumina membranes by selectively removing the anion-included layer (75% of the cell width) and heating phosphoric acid films formed at high voltages of about 200 V. Although this technique avoids the volume change caused by the decomposition of anions, it cannot be applied to the fabrication of membranes with small pore sizes using oxalic acid films, where the thickness ratio of the anion-free layer is only 10% and, thus, cannot produce mechanically resistant membranes. If appropriate α-alumina membranes with high chemical resistance can be manufactured, they can be used for filtration in aggressive environments and recycled. In addition, the fabrication of α-alumina membranes with an extensive range of pore diameters is also essential.

We successfully fabricated nanoporous and single-phase α-alumina membranes with pore diameters tunable over a wide range of approximately 30–350 nm by optimizing the conditions for anodizing and the subsequent detachment and heat treatment [92,93,100]. The electrolytically stripped film was chemically dissolved with phosphoric acid to remove the remaining barrier layer at the bottom of the pore, followed by through-hole treatment. The film was then constrained between ceramic plates and crystallized into α-alumina by heating at a predetermined temperature rise rate. The oxalic acid and phosphoric acid films were heat-treated at 1250 °C and 1400 °C, respectively, for 4 h. SEM images of the surfaces of the membranes before heating (a–d) and after heating (e–h), which is α-alumina, are shown in Figure 18 [92,93]. Sintering progressed on both the front and back surfaces, and the membrane’s fine irregularities were reduced and smoothed when compared to before heating.

The pore diameter increased, and the cell diameter shrunk upon crystallization to α-alumina by approximately 20% and 3%, respectively, by the 23% volume shrinkage resulting from the change in density associated with the transformation from the amorphous state to α-alumina. Nonetheless, without thermal deformation, flat α-alumina membranes with diameters of 25 mm and thicknesses of 50 μm were obtained. The α-alumina membranes demonstrated excellent chemical resistance in a variety of concentrated acidic and alkaline solutions, as well as when subjected to high-temperature steam under pressure. The Young’s modulus and hardness of the single-phase α-alumina membranes formed by heat treatment were notably decreased compared to the corresponding amorphous membranes, presumably because of the nodular crystallite structure of the cell walls and the substantial increase in porosity. Furthermore, when used for filtration, the α-alumina membrane demonstrated a higher level of flux than the commercial ceramic membrane [92].

## 12. Conclusions

The author described her research on porous anodic films formed on aluminum, with a focus on the effects of the electric field strength and anion incorporation on the nanostructures of the films. The content and depth of anion incorporation in the anodic films increased linearly with log *j* (i.e., *E*). From the results of the cell parameter measurements, it was deduced that the barrier layer thickness and cell diameter were proportional to the voltage and inversely proportional to *E*, whereas the pore diameter was proportional to the voltage and inversely proportional to the square of *E*. Consequently, the pore size depends more strongly on the current density than on the other cell parameters. Thus, the electric field is a key factor that determines the nanostructure and properties of anodic films, including self-organization.

The cause of the formation of the small initial pores on the film surface and the radially branched nanopores in cell walls of major pores is the same: low electric field strength. Special emphasis was placed on showing the real image of the film, including defects/irregularities such as spherical voids and side holes, as well as radial nanobranches of pores, which have been conventionally considered atypical structures. The formation of radial pore branching and other atypical structures is not unique to special electrolyte but is a universal phenomenon that occurs in most electrolytes during anodizing.

It is important to explain the controlling factors of the film structure in a universal and unified manner for films formed in the different types of anodizing electrolytes, which would be the electric field strength during anodic film growth. Based on the author’s long experience of joint research with companies, a deep fundamental knowledge of the controlling factors of anodic oxide films is of the utmost importance for not only scientific basic research but, also, industrial applied research and development. The findings presented here are very simple, and it would be a great pleasure if they led to further clarification of the mechanisms of anodic film formation, the development of new applications, and the reduction of the economic costs of corrosion-resistant film formation in the future.

## 13. Patents

Ono, S. et al., Film Formation Method and Metallic Material. Japan Patent 6753899, 2020. Ono, S. et al., The aluminum member and manufacturing method thereof. Japan Patent 6604703, 2019. Ono, S. et al., Manufacturing method for porous materials. Japan Patent 5611618, 2014. Ono, S. et al., Process for producing an aluminum electrode material for an electrolytic capacitor and aluminum of the aluminum material for an electrolytic capacitor electrode having excellent etching properties. Japan Patent 5373745, 2013. Ono, S. et al., The aluminum material for an electrolytic capacitor electrode and its manufacturing method, an electrode material for aluminum electrolytic capacitors, and aluminum electrolytic capacitor. Japan Patent 5329686, 2013. Ono, S. et al., An electrolytic capacitor aluminum electrode material for etching aluminum substrates using the same of. Japan Patent 5094172, 2012. S. Ono et al., To obtain an excellent etching characteristics for an electrolytic capacitor electrode aluminum material and its manufacturing method, an electrode material for aluminum electrolytic capacitors and ar. Japan Patent 4763363, 2011. S. Ono et al., The aluminum or aluminum alloy used for the vacuum device and its components, method of surface treatment, the vacuum device and its components. Japan Patent 3917966, 2007. Ono, S. et al., Method for formation of anode oxide film. Japan Patent 3963763, 2007.

## Figures and Tables

**Figure 1 molecules-26-07270-f001:**
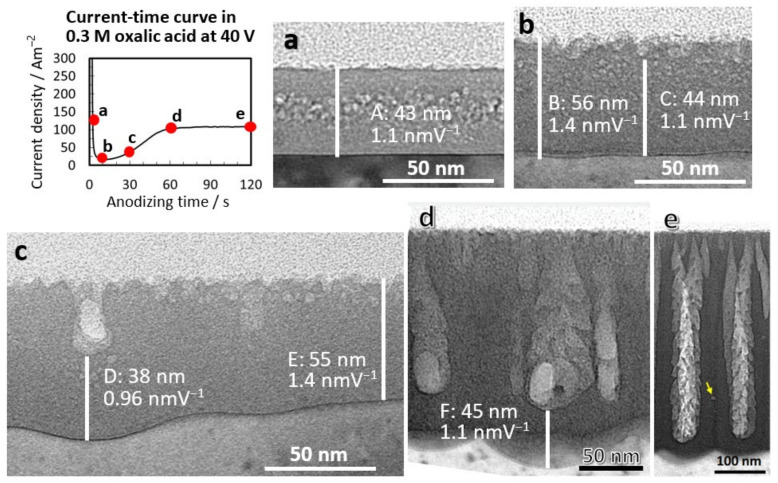
TEM images of vertical cross-sections of anodic films formed in (**a**–**c**) 0.3-mol·dm^−3^ oxalic acid at 40 V for (**a**) 2.5 s, (**b**) 10 s, (**c**) 30 s, (**d**) 60 s, and (**e**) 120 s and the corresponding current time curve. The anodizing ratios measured by the thickness of the barrier layer of each oxide film are indicated in TEM images. The yellow arrow indicates a spherical void at triple cell junction. Figures from Ono and Asoh [16].

**Figure 2 molecules-26-07270-f002:**
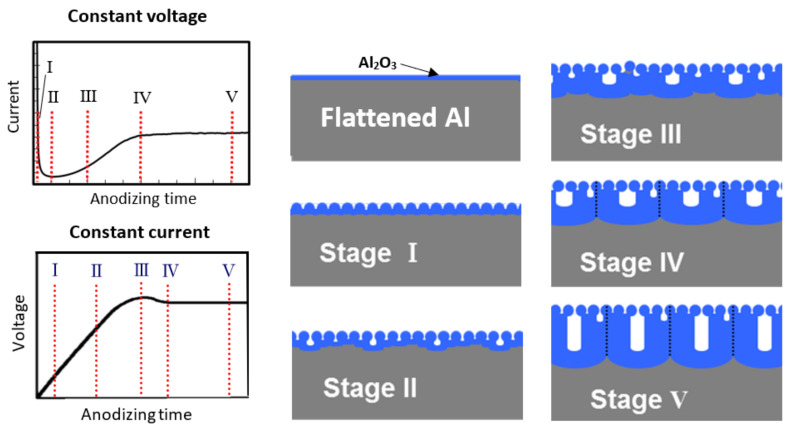
Schematic representation of the growth process of porous anodic film at different stages. Each stage was indicated in the current–time curve at constant voltage anodizing and the voltage–time curve at constant current anodizing. Some figures modified from Ono and Asoh [36].

**Figure 3 molecules-26-07270-f003:**
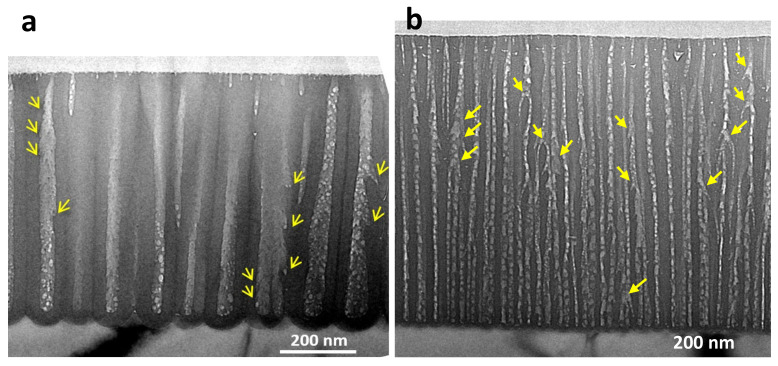
TEM images of cross-sections of anodic films formed in (**a**) 0.3-mol dm^−3^ oxalic acid at 40 V for 3 min and (**b**) 1.5-mol dm^−3^ sulfuric acid at 20 V for 1 min. Arrows indicate radial pore branching. Figures from Ono and Asoh [36].

**Figure 4 molecules-26-07270-f004:**
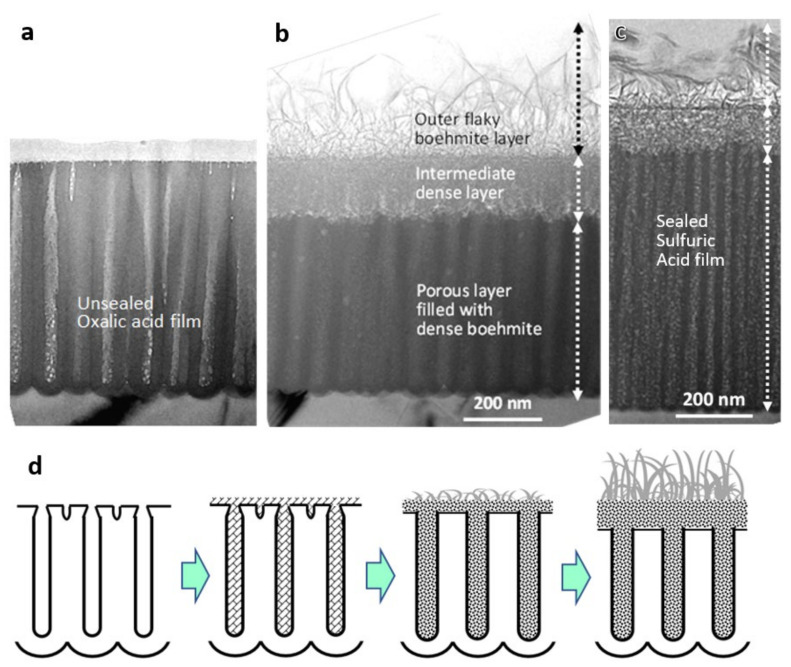
TEM images of cross-sections of anodic films formed in (**a**,**b**) 0.3-mol dm^−3^ oxalic acid at 40 V for 3 min and (**c**) 1.5-mol dm^−3^ sulfuric acid at 20 V for 1 min (**a**) before and (**b**,**c**) after sealing in boiling water for 10 min. (**d**) Schematic of the sealing process in hot water. Figures from Ono and Asoh [36].

**Figure 5 molecules-26-07270-f005:**
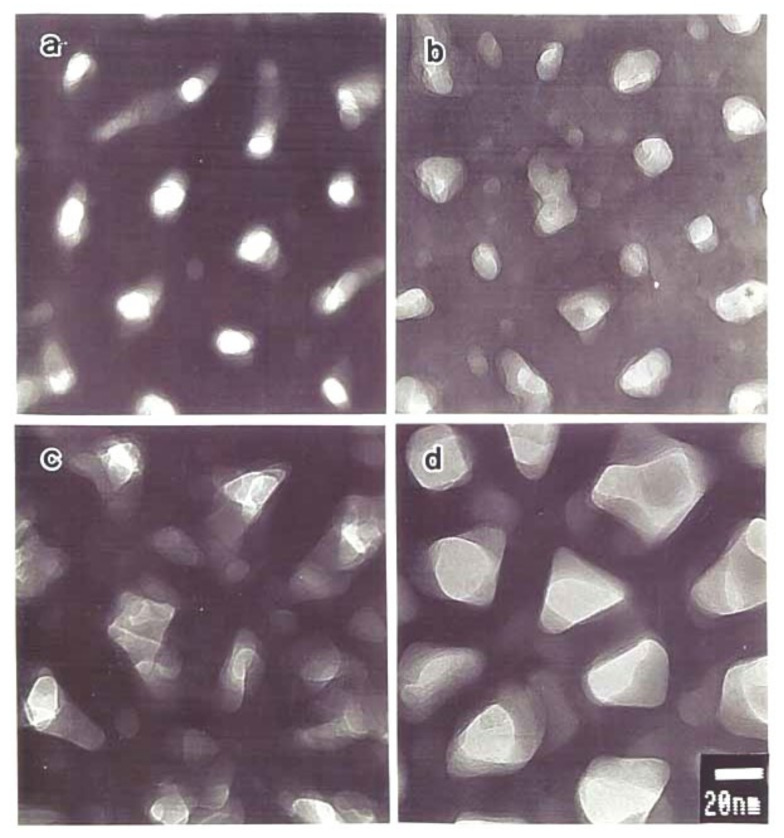
TEM images of stripped anodic films formed at 20 V in (**a**) 1-mol dm^−3^ sulfuric acid at 20 °C, (**b**) 0.3-mol dm^−3^ oxalic acid at 30 °C, (**c**) 0.3-mol dm^−3^ chromic acid at 40 °C, and (**d**) 0.4-mol dm^−3^ phosphoric acid at 25 °C. Figures from Ono et al. [59].

**Figure 6 molecules-26-07270-f006:**
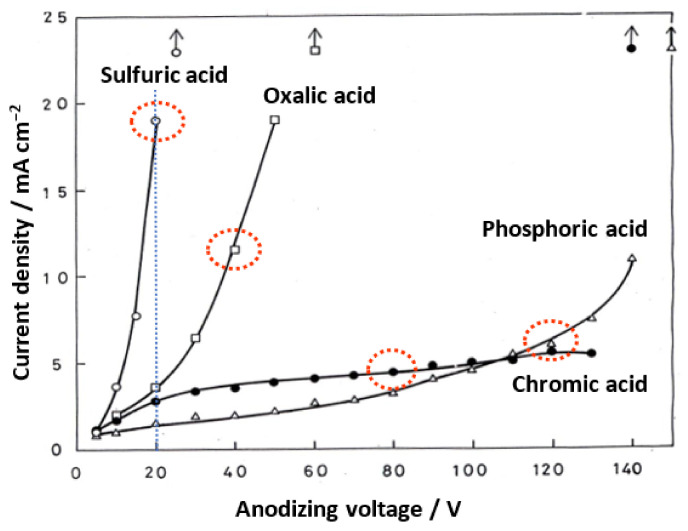
Change in the steady-state current density with the voltage, where anodizing was performed at constant voltage in 1-mol dm^−3^ sulfuric acid, 0.3-mol dm^−3^ oxalic acid, 0.3-mol dm^−3^ chromic acid, and 0.4-mol dm^−3^ phosphoric acid. The red dotted ellipses indicate the commonly used voltages for each electrolyte. Parts of this figure were modified from Ono et al. [61].

**Figure 7 molecules-26-07270-f007:**
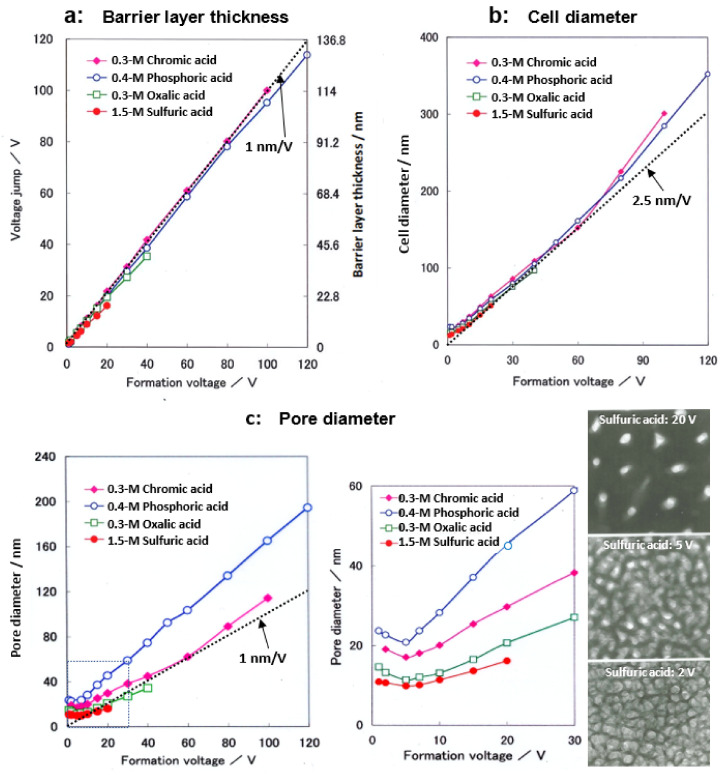
Changes in (**a**) the barrier layer thickness, (**b**) cell diameter, and (**c**) pore diameter with the formation voltage of the films formed in 1.5-mol dm^−3^ sulfuric acid, 0.3-mol dm^−3^ oxalic acid, 0.3-mol dm^−3^ chromic acid, and 0.4-mol dm^−3^ phosphoric acid. TEM images of stripped films formed in sulfuric acid at 20 V, 5 V, and 2 V are attached. Part of this figure is from Ono and Masuko [14,15]. The results from 40–120 V are newly determined.

**Figure 8 molecules-26-07270-f008:**
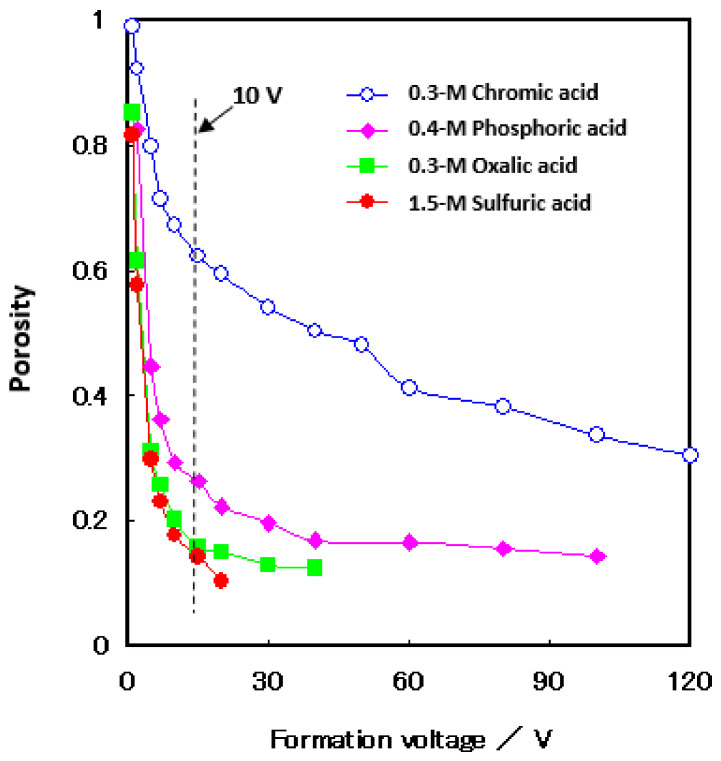
Changes in porosity with the formation voltage of the films formed in 1.5-mol dm^−3^ sulfuric acid, 0.3-mol dm^−3^ oxalic acid, 0.3-mol dm^−3^ chromic acid, and 0.4-mol dm^−3^ phosphoric acid. The porosity remarkably increased below 10 V in the formation voltage. Pars of the data are from Ono et al. [14,61].

**Figure 9 molecules-26-07270-f009:**
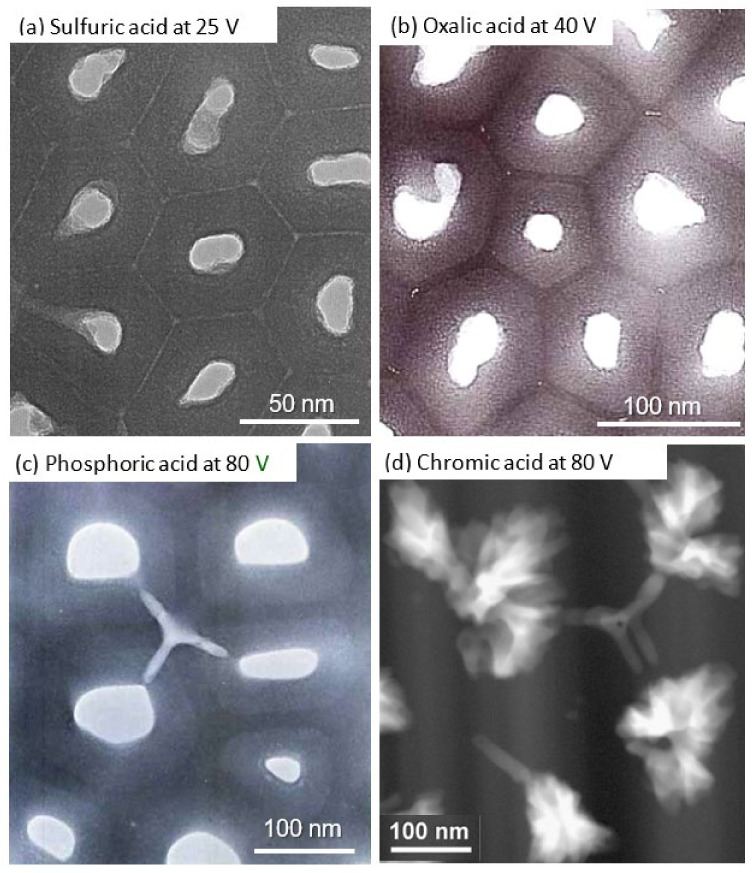
TEM images of horizontal cross-sections of the films formed at conventional anodizing conditions in (**a**) 1-mol dm^−3^ sulfuric acid at 25 V [69], (**b**) 0.3-mol dm^−3^ oxalic acid at 40 V [60], (**c**) 0.4-mol dm^−3^ phosphoric acid at 80 V [13], and (**d**) 0.3-mol dm^−3^ chromic acid at 80 V [16].

**Figure 10 molecules-26-07270-f010:**
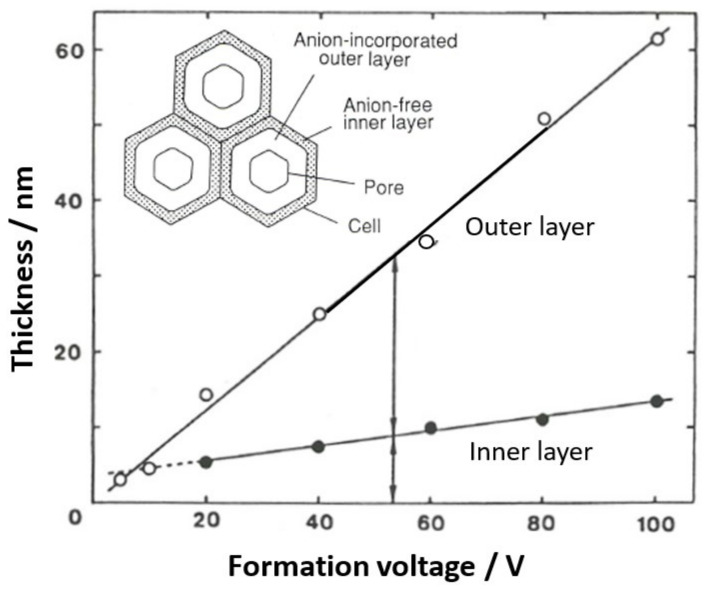
Change in the thickness of the anion-incorporated outer layer and anion-free inner layer of the figure. Displayed by mol dm^−3^ phosphoric acid with the formation voltage. The figure is from Ono and Masuko [38].

**Figure 11 molecules-26-07270-f011:**
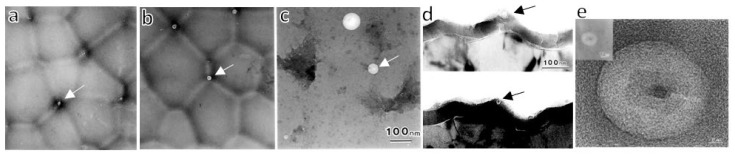
TEM images of stripped barrier films formed on uneven Al substrate that has hexagonal cell topography. Anodizing was conducted in 0.1-mol dm^−3^ ammonium borate solution at 50 A m^−2^ up to (**a**) 10 V, (**b**) 30 V, (**c**) 120 V, (**d**) 80 V, and (**e**) 40 V. (**d**) Cross-sections of the spherical voids formed on protruded substrate. (**e**) A spherical void formed on the substrate protrusion and lattice images of the inner particle. Arrows indicate spherical voids. Part of the figures are from Ono et al. [76].

**Figure 12 molecules-26-07270-f012:**
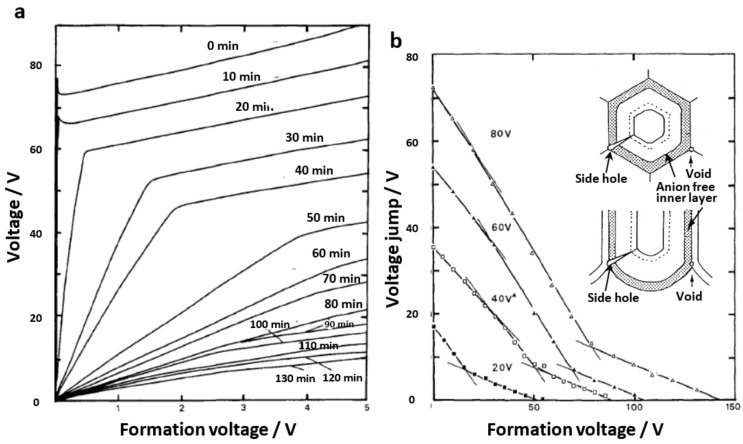
(**a**) Voltage–time curves during re-anodizing in neutral sodium borate solution at 1 A m^−2^ of the films formed in 0.4-mol dm^−3^ phosphoric acid at 80 V. Anodic films were dipped in 2-mol dm^−3^ sulfuric acid at 50 °C for chemical dissolution for each 10 min up to 130 min. (**b**) Change in the voltage jump that corresponds to the barrier layer thickness during re-anodizing of the films after chemical dissolution in sulfuric acid. Anodic films were formed in 0.4-mol dm^−3^ phosphoric acid at different voltages. Inset indicates defects in the film causing a delay of the voltage increase, as shown in (**a**). Parts of the figure are from Ono and Masuko [73].

**Figure 13 molecules-26-07270-f013:**
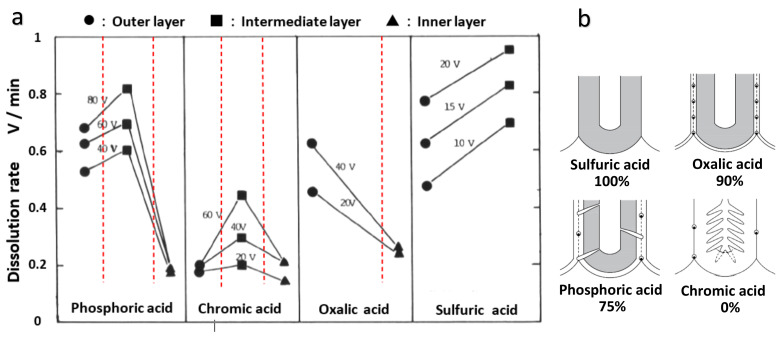
(**a**) Change in the dissolution rate of the outer, intermediate, and inner layers of the barrier layers of the films formed in four typical electrolytes. (**b**) A schematic diagram of the defects in the four types of films, where the hatched areas are anion-incorporated oxides [12,13,16,60]. The fraction of the anion-incorporated layer of each film is indicated. Parts of the figure are from Ono and Masuko [73].

**Figure 14 molecules-26-07270-f014:**
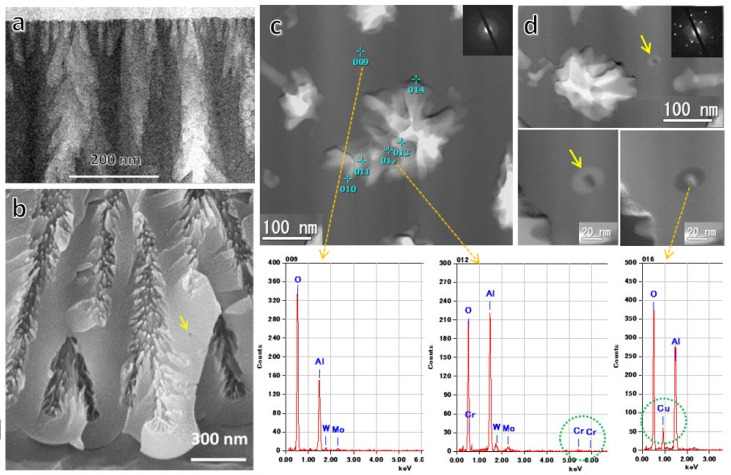
(**a**,**c**,**d**) TEM images, energy-dispersive X-ray spectroscopy, and ED microanalysis of the horizontal cross-sections of anodic films formed in 0.3-mol dm^−3^ chromic acid at 80 V. (**b**) SEM image of the fracture section of the film formed in chromic acid at 120 V. Arrows indicate a spherical void, including a particle. W and Mo are derived from the sample preparation process. From Ono and Asoh [16].

**Figure 15 molecules-26-07270-f015:**
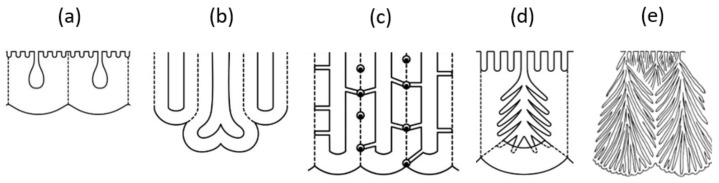
A schematic diagram showing various atypical pore structures. (**a**) Small initial pores on the top surface, (**b**) branching main pore, (**c**) straight side holes across the cell wall and spherical voids with Cu-rich particles at triple-cell junctions, (**d**) radially branched nanopores within the main pores, and (**e**) a branching colony structure comprising a bundle of radially branched main pores. From Ono and Asoh [16].

**Figure 16 molecules-26-07270-f016:**
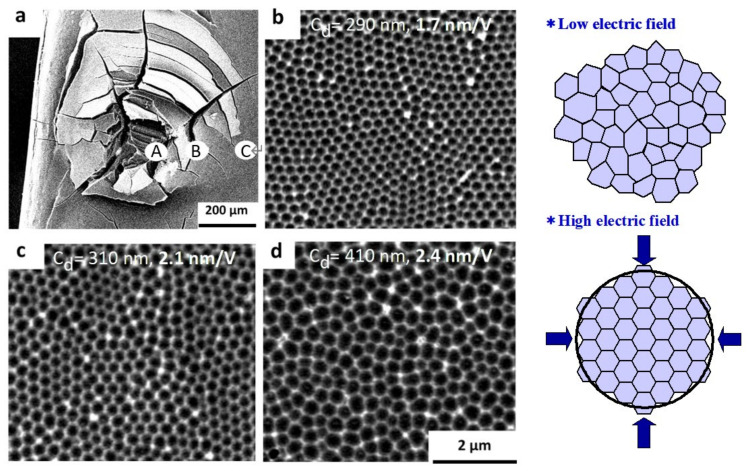
(**a**) SEM image of a protruding part with numerous cracks in the thickened anodic oxide film caused by burning. The protruding part is divided into three regions: (**a**) the center, (**b**) the middle, and (**c**) the outside. (**b**–**d**) SEM images of the respective substrate surfaces of the three regions corresponding to A, B, and C after the removal of anodic films. A schematic model of the cell arrangement at low and high electric fields is shown. From Ono et al. [61,63].

**Figure 17 molecules-26-07270-f017:**
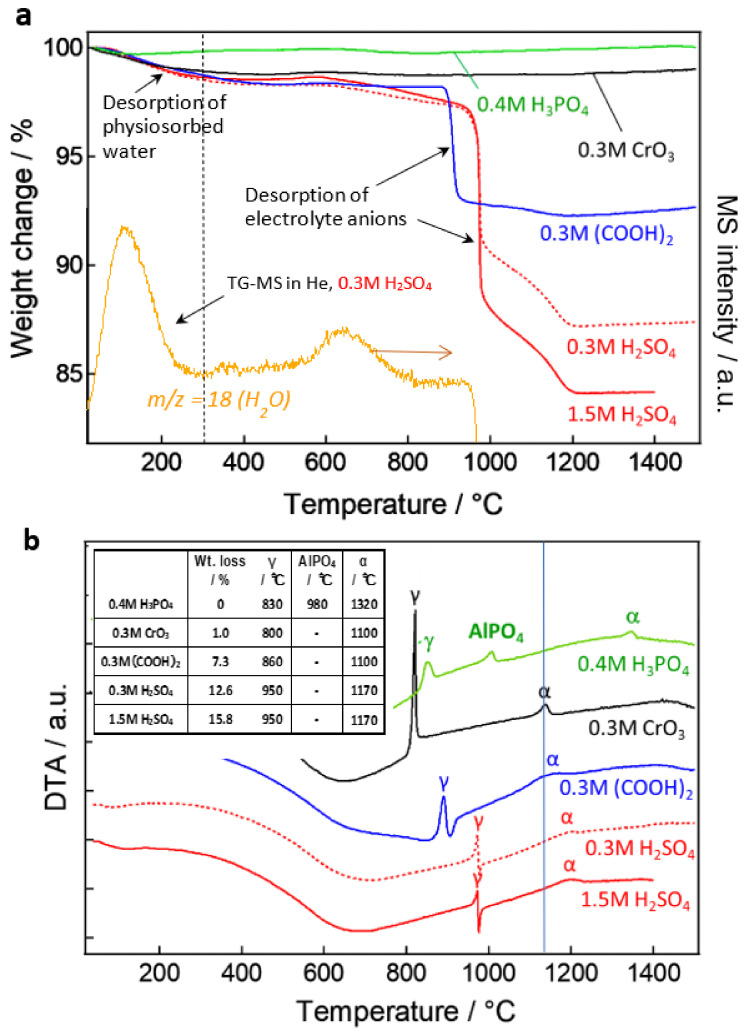
(**a**) TG curves in an air of anodic films formed in different anodizing conditions. A TG-MS curve in He of an anodic film formed in 0.3-mol dm^−3^ sulfuric acid measuring H_2_O is also shown. The heating rate was 10 °C/min in both cases. (**b**) DTA curves in an air of anodic films formed in different anodizing conditions. Inset shows the phase transition temperature and total weight loss other than pysisorbed water of different anodic films. Parts of the figure are from Masuda et al. and Hashimoto et al. [92,93,94].

**Figure 18 molecules-26-07270-f018:**
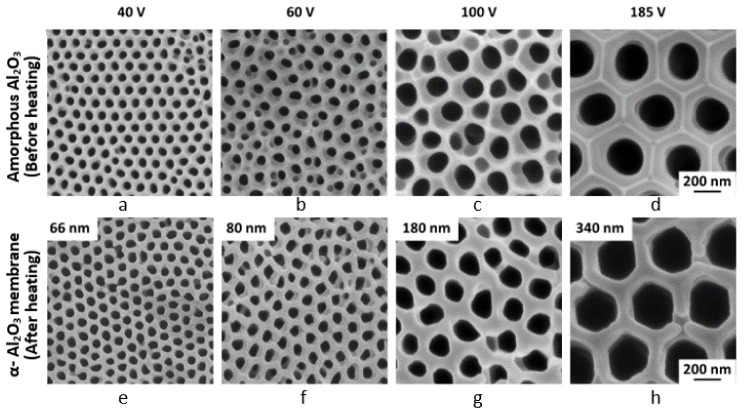
SEM images of the surfaces of anodic porous alumina formed at (**a**,**c**) 40 V, (**b**,**f**) 60 V, (**c**,**g**) 100 V, and (**d**,**h**) 185 V before heating (**a**–**d**) and after heating to α-alumina (**e**–**h**). From Masuda et al. [92,93].

**Table 1 molecules-26-07270-t001:** Changes in the current density, thickness fraction of the anion-incorporated outer layer, and PO_4_ content of the films formed in 0.4-mol dm^−3^ phosphoric acid with the formation voltage. From Ono et al. [38].

Formation Voltage/V	Current Density/A m^−2^	Fraction of Outer Layer	PO_4_ Content/wt.%
2	8	-	5.6
5	9	-	1.8
10	11	-	2.2
20	15	0.62	3.7
40	21	0.69	5.4
60	27	0.71	6.3
80	32	0.77	6.9
100	45	0.78	7.4

**Table 2 molecules-26-07270-t002:** Fraction of coordination numbers obtained by NMR measurements and the chemical compositions of anodic films formed in various anodizing conditions. Part of the data is from Hashimoto et al. [91].

Anodizing Condition, Composition ↓/Coordination Number →	^[4]^Al/%	^[5]^Al/%	^[6]^Al/%	Average
0.3-M Chromic acid, 40 °C, 80 V/Al_2_O_3_	41.3	52.0	6.7	4.65
0.4-M Phosphoric acid, 25 °C, 120 V/Al_2_O_2.884_(PO_4_)_0.077_	41.6	54.2	4.2	4.63
0.3-M Oxalic acid, 40 °C, 40 V/Al_2_O_2.913_(C_2_O_4_)_0.069_(OH)_0.035_	41.9	50.7	7.4	4.66
0.3-M Sulfuric acid, 20 °C, 25 V/Al_2_O_2.846_(SO_4_)_0.125_(OH)_0.059_	39.1	53.2	7.7	4.69
1.5-M Sulfuric acid, 20 °C, 100 Am^−2^/Al_2_O_2.801_(SO_4_)_0.174_(OH)_0.049_	32.8	56.3	10.9	4.78

## Data Availability

The raw/processed data required to reproduce these findings cannot be shared at this time due to technical or time limitations.

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
