# Peer review of "Nanostructure Analysis of Anodic Films Formed on Aluminum-Focusing on the Effects of Electric Field Strength and Electrolyte Anions"

_molecules, 2021, doi:10.3390/molecules26237270_

Round 1

Reviewer 1 Report

The authors have written the manuscript with interesting and valuable results. They proposed a new perspective at the pores of the anodic oxide film for the electric field strength decreases of the pores of the anodic oxide film grow in a radially branching materials. Authers aimed to clarify the mechanism of hot water sealing in particular, paying attention to the temporal change during the reaction in the sealing process, differences in the local structure of the film in the sealing process. I have read the present manuscript with interest. I have, however, found that this work can be considered for publication.

Author Response

Answers for Reviewers' comments:

The authors are deeply grateful to the editors and reviewers for their courteous review and appropriate comments. They have greatly helped improve this paper. All changes to the dissertation text are displayed in yellow marker.

Reviewer 1

The author is deeply grateful to the reviewer for his/her deep understanding and positive evaluation of my manuscript.

Reviewer 2 Report

Submitted article entitled “Nanostructure Analysis of Anodic Films Formed on Aluminium – Focusing on the Effects of Electric Field Strength and Electrolyte Anions” by Sachiko Ono concludes and summarizes research results that the Author and her co-workers have obtained during 40 years of their work on aluminium anodization. Therefore, it is not surprising that 43% of cited references stands for author’s work and 58% of all cited reports is not referring to research done in 2 recent decades (only 13% from last 5 years). Presented findings, however, are set into the context of using modern and sophisticated techniques (such as electron microscopy) to characterize structural features of generated alumina films and, ultimately, to understand mechanisms such films grow through. Thus, presented manuscript explores in details the impact of applied conditions (with special focus on electric filed strength) and electrolyte composition on morphology of alumina films grown in anodization process. It is well written and provides comprehensive knowledge of undertaken subject, which may draw attention of audience interested in researching metal anodizing.

However, in my opinion, 43% of self-citing is inappropriate ratio and this issue should be addressed before considering this manuscript for publishing in the journal. I would like to emphasize that work which has been done by the author on metal anodizing deserves appreciation, nonetheless, excessive self-citating in submitted manuscript is not in line with publishing standards. I recommend to put presented findings on Al anodization into broader perspective by including results of recent studies reported by different researchers in the field of Al anodization. There are numerous reports examining ion incorporation, growth mechanisms or pores’ sealing during Al anodizing elaborated within the last decade that could be referred in this review to provide a basis of comparison for a reader. 

As submitted manuscript gathers findings presented previously by the author, original source, as well as a note on reprint permission for the reused graphics have to be included in figures’ captions. That would also enable easier and more convenient navigation for a reader who would like to know more details on research given figure refers to.

Figure mentioned in subsection 2.3 should refer to figure 3 not 2 (numbering is wrong).              
Figure mentioned in subsection 4.2 should refer to figure 6 not 5 (numbering is wrong).

After addressing above-mentioned issues, I think the manuscript will be suitable for publishing in Molecules journal.

Author Response

Reviewer 2

Submitted article entitled “Nanostructure Analysis of Anodic Films Formed on Aluminium – Focusing on the Effects of Electric Field Strength and Electrolyte Anions” by Sachiko Ono concludes and summarizes research results that the Author and her co-workers have obtained during 40 years of their work on aluminium anodization. Therefore, it is not surprising that 43% of cited references stands for author’s work and 58% of all cited reports is not referring to research done in 2 recent decades (only 13% from last 5 years). Presented findings, however, are set into the context of using modern and sophisticated techniques (such as electron microscopy) to characterize structural features of generated alumina films and, ultimately, to understand mechanisms such films grow through. Thus, presented manuscript explores in details the impact of applied conditions (with special focus on electric filed strength) and electrolyte composition on morphology of alumina films grown in anodization process. It is well written and provides comprehensive knowledge of undertaken subject, which may draw attention of audience interested in researching metal anodizing. 

However, in my opinion, 43% of self-citing is inappropriate ratio and this issue should be addressed before considering this manuscript for publishing in the journal. I would like to emphasize that work which has been done by the author on metal anodizing deserves appreciation, nonetheless, excessive self-citating in submitted manuscript is not in line with publishing standards. I recommend to put presented findings on Al anodization into broader perspective by including results of recent studies reported by different researchers in the field of Al anodization. There are numerous reports examining ion incorporation, growth mechanisms or pores’ sealing during Al anodizing elaborated within the last decade that could be referred in this review to provide a basis of comparison for a reader.

The author would like to thank the reviewer for his/her understanding and helpful advice in improving the content of the paper. I have made the greatest effort to cite new research. Currently, my self-citation rate has dropped to 33%, and the number of cited papers published within the last decade (since 2011) is 29%. I would appreciate it if you would acknowledge my efforts to improve the current situation, since many of the papers that are appropriate references for this paper are relatively classical. I have more respect for proven results than for sophisticated but conceptual results. I revised the manuscript accordingly.

As submitted manuscript gathers findings presented previously by the author, original source, as well as a note on reprint permission for the reused graphics have to be included in figures’ captions. That would also enable easier and more convenient navigation for a reader who would like to know more details on research given figure refers to.

I well understand the suggestion. I added reference number to all figures and table though some of data used for figures in this paper are original (not published yet). Concerning the reprint permission for the reused graphics, because of contacting the publisher, I was taught that all I had to do was write the source of the citation and I didn't need permission to reprint it in my own work.

Figure mentioned in subsection 2.3 should refer to figure 3 not 2 (numbering is wrong).              
Figure mentioned in subsection 4.2 should refer to figure 6 not 5 (numbering is wrong).

Thank you for pointing this out. I corrected wrong writing.

After addressing above-mentioned issues, I think the manuscript will be suitable for publishing in Molecules journal.

Thank you for all your efforts to improve my manuscript.

Reviewer 3 Report

Nanostructure Analysis of Anodic Films Formed on Aluminum Focusing on the Effects of Electric Field Strength and Electrolyte Anions review manuscript explained effects of electrolyte anion incorporation and electric field strength on the nanostructure of anodic oxide films. I recommended it for publication after minor revisions. Here are the comments:

  1. Please in the Introduction part gave several examples of anodic films formed on aluminum and present the application of these anode films.
  2. Why is this review important for future investigation? Please note the reasons in the Introduction part. E.g. I saw the next sentence: ĘĽĘĽHowever, I hope that the results of the basic analyses presented here will lead to the reader's research and technology being further deepened and developed. ĘĽĘĽ Please detailed explain this part.
  3. Please add an appropriate reference of 3.3. Sealing mechanism section for each paragraph. Please do the same thing for 4.2. Current density – voltage diagram section. Please revise all manuscripts and add the appropriate reference.
  4. In the review manuscript is noted that:

ĘĽĘĽThe thickness of the barrier layer was not the same for the four electrolytes but was larger for sulfuric acid, oxalic acid, phosphoric acid, and chromic acid in that order. ĘĽĘĽ

ĘĽĘĽ Sulfuric acid, oxalic acid, chromic acid, and phosphoric acid have larger pore sizes at the same voltage, in that order. ĘĽĘĽ

ĘĽĘĽAccording to the difference in dissolution rate, the film formed in phosphoric acid has three layers, the film formed in oxalic acid has two layers, the film formed in chromic acid has three layers, and the film formed in sulfuric acid has two layers. ĘĽĘĽ Etc.

Which electrolytes are the best choice for optimal results?

  1. Please add concrete facts about controlling factors of the film structure in the Conclusion part. Also, please, explain why this review is important for the reader's research.

Author Response

Reviewer 3

  1. Please in the Introduction part gave several examples of anodic films formed on aluminum and present the application of these anode films.

I inserted several examples of anodic alumina films and the application in the Introduction part as follows.

“Specific applications that take advantage of its corrosion resistance and decorative properties include surface treatment of aluminum sashes, housings for electrical equipment, aluminum alloys for automobile engines, and the inner surface of vacuum chambers. In addition to filters, catalyst supports, and photonic crystals, the nano-porous structure of an anodic film can be used as a template for the creation of nanomaterials for electronic, magnetic, and optical devices, including batteries. In recent years, the creation of anti-reflective coatings has become a practical application. The dielectric properties of barrier-type anodic alumina are widely used as dielectrics in electrolytic capacitors and as gate insulators.”

  1. Why is this review important for future investigation? Please note the reasons in the Introduction part. E.g. I saw the next sentence: ĘĽĘĽHowever, I hope that the results of the basic analyses presented here will lead to the reader's research and technology being further deepened and developed. ĘĽĘĽ Please detailed explain this part.

I have explained details for this subject as follows:

“However, the basic mechanism of anodic film growth is not fully elucidated, even though numerous studies on the technology directly necessary for its practical use, such as the fabrication of highly ordered films, have been undertaken [11]. The creation of oxide films with new functions cannot be achieved without an understanding of the basic formation mechanism, including the effects of electric fields on film nanostructure and anion incorporation. For example, to develop methods to produce high-quality anodic oxide films with lower energy costs, a deeper understanding of the mechanisms of film growth is needed, and the currently accepted knowledge may not be sufficient.”

  1. Please add an appropriate reference of 3.3. Sealing mechanism section for each paragraph. Please do the same thing for 4.2. Current density – voltage diagram section. Please revise all manuscripts and add the appropriate reference.

I have added references for all manuscript though some of data used for the figures in this paper are original (not published yet). I hope this revision is appropriate.

  1. In the review manuscript is noted that:

ĘĽĘĽThe thickness of the barrier layer was not the same for the four electrolytes but was larger for sulfuric acid, oxalic acid, phosphoric acid, and chromic acid in that order. ĘĽĘĽ

ĘĽĘĽ Sulfuric acid, oxalic acid, chromic acid, and phosphoric acid have larger pore sizes at the same voltage, in that order. ĘĽĘĽ

ĘĽĘĽAccording to the difference in dissolution rate, the film formed in phosphoric acid has three layers, the film formed in oxalic acid has two layers, the film formed in chromic acid has three layers, and the film formed in sulfuric acid has two layers. ĘĽĘĽ Etc.

Which electrolytes are the best choice for optimal results?

Thank you for asking this question. I have added following sentences accordingly. Each film has its own characteristics, and people can choose the appropriate electrolyte for their own purposes. For this reason, four types of electrolytes have been used industrially to date.

“The characteristics of the four typical anodic oxide films formed in each type of electrolyte are as follows. Sulfuric acid, which is inexpensive and the most widely used electrolyte in the anodizing industry, produces a transparent film. Sulfuric acid films are used not only for corrosion resistance and decorative purposes but also for templates when a smaller size is required due to its low formation voltage of around 20 V, which also lowers energy costs. Oxalic acid film has a formation voltage of around 40 V and is used when a higher corrosion resistance is required. The film is transparent but slightly yellow. Chromic acid film is an opaque grayish-white in color and has a ceramic-like appearance. Anodizing process is carried out at approximately 60 V or higher and it is mainly used for Al alloys for aircraft. Phosphoric acid film is formed at relatively high voltages of 80 V or more, has a whitish appearance, and is characteristically difficult to hydrate. It is also used as a base for painting.”

  1. Please add concrete facts about controlling factors of the film structure in the Conclusion part. Also, please, explain why this review is important for the reader's research.

I have added following sentences accordingly in the conclusion part. Thank you for your advice.

“The content and depth of anion incorporation in the anodic films increased linearly with log j (i.e., E). From the results of cell parameter measurements, it was deduced that barrier layer thickness and cell diameter were proportional to voltage and inversely proportional to E, whereas pore diameter was proportional to voltage and inversely proportional to the square of E. Consequently, the pore size depends more strongly on the current density than on other cell parameters. Thus, the electric field is a key factor that determines the nanostructure and properties of anodic films, including self-organization.

The cause of the formation of the small initial pores on the film surface and the radially branched nanopores is the same: low electric field strength.”

I have inserted the reasons why this review is important in the introduction, as noted in my response to question 2. In addition, the following text has been added to the conclusion section.

“It is important to explain the controlling factors of the film structure in a universal and unified manner for films formed in the different types of anodizing electrolytes, which would be the electric field strength during anodic film growth. Based on the author’s long experience of joint research with companies, a deep fundamental knowledge of the controlling factors of anodic oxide films is of the utmost importance for not only scientific basic research but also industrial applied research and development.”

I would like to thank you for all your efforts which helped me a lot in improving my manuscript.

Round 2

Reviewer 2 Report

Author's effort to improve the manuscript should be acknowladged. The number of self-citing has been lowered and some of more current research have been included. It is clear that adding current but irrelevant findings which are not within the scope of this review would enhance the manuscript in no way, hance, such practice should be avoided. Nonetheless, Author adressed all issues highlighted previously and therefore the manuscript can be proceeded for publication in the journal.